# Acute Influenza A virus outbreak in an enzootic infected sow herd: Impact on viral dynamics, genetic and antigenic variability and effect of maternally derived antibodies and vaccination

Pia Ryt-Hansen[1]*, Anders Gorm Pedersen[2], Inge Larsen[3], Jesper Schak Krog[1], Charlotte Sonne Kristensen[4], Lars Erik Larsen[1,3]

1 National Veterinary Institute, Technical University of Denmark, Kongens Lyngby, Denmark, 2 Department of Health Technology, Section for Bioinformatics, Technical University of Denmark, Kongens Lyngby, Denmark, 3 University of Copenhagen, Dpt. of Veterinary and Animal Sciences, Frederiksberg C, Denmark, 4 SEGES, Danish Pig Research Centre, Aarhus N, Denmark

* pryt@vet.dtu.dk

**Data Availability Statement:** All sequencing files are available from the NCBI Genbank database

## Abstract

Influenza A virus (IAV) is a highly contagious pathogen in pigs. Swine IAV (swIAV) infection causes respiratory disease and is thereby a challenge for animal health, animal welfare and the production economy. In Europe, the most widespread strategy for controlling swIAV is implementation of sow vaccination programs, to secure delivery of protective maternally derived antibodies (MDAs) to the newborn piglets. In this study we report a unique case, where a persistently swIAV (A/sw/Denmark/P5U4/2016(H1N1)) infected herd experienced an acute outbreak with a new swIAV subtype (A/sw/Denmark/HB4280U1/2017(H1N2)) and subsequently decided to implement a mass sow vaccination program. Clinical registrations, nasal swabs and blood samples were collected from four different batches of pigs before and after vaccination. Virus isolation, sequencing of the virus strain and hemagglutinin inhibition (HI) tests were performed on samples collected before and during the outbreak and after implementation of mass sow vaccination. After implementation of the sow mass vaccination, the time of infection was delayed and the viral load significantly decreased. An increased number of pigs, however, tested positive at two consecutive sampling times indicating prolonged shedding. In addition, a significantly smaller proportion of the 10–12 weeks old pigs were seropositive by the end of the study, indicating an impaired induction of antibodies against swIAV in the presence of MDAs. Sequencing of the herd strains revealed major differences in the hemagglutinin gene of the strain isolated before- and during the acute outbreak despite that, the two strains belonged to the same HA lineage. The HI tests confirmed a limited degree of cross-reaction between the two strains. Furthermore, the sequencing results of the hemagglutinin gene obtained before and after implementation of mass sow vaccination revealed an increased substitution rate and an increase in positively selected sites in the globular head of the hemagglutinin after vaccination.

accession numbers: MN410726-MN410785 and MN410796-MN410883. The remaining relevant data are within the manuscript and its Supporting Information files.

**Funding:** The authors received no specific funding for this work.

**Competing interests:** The authors have declared that no competing interests exist.

## Introduction

Influenza A virus (IAV) in swine (swIAV) is an enzootic virus of swine herds globally. SwIAV infects the cells of the respiratory tract, inducing clinical signs of respiratory disease and fever [1–3]. Additionally, IAV impairs the immune system, making the infected pig more susceptible to other pathogens [4–6]. During the last 10–20 years, the pig industry has undergone profound structural changes resulting in a significant increase in herd size and a continuous movement of pigs between production units. These changes has altered the dynamics of swIAV from an epizootic disease that resolved in a few weeks to a more enzootic situation with persistent circulation of the virus in same herds for years due to the continuous exposure of naïve piglets [7–15]. These changes have emphasized that there is a need for effective control measures at the herd level, and have resulted in a marked increase in the sales of IAV vaccines. One of the most used vaccines on the European market is Respiporc FLU3, which is an inactivated, adjuvanted, whole virus trivalent vaccine including the subtypes; Bakum/IDT1769/2003 (H3N2), Haselünne/IDT2617/2003 (H1N1) and Bakum/1832/2000 (H1N2) [16]. This vaccine is currently used in Danish herds to control infections with H1av- like viruses including H1, which is the most prevalent subtype found in Denmark [17]. In general, the H1av-like viruses circulating in Denmark share a high level of genetic identity to the H1av-like viruses circulating in the rest of Europe [18].

As pigs have an impermeable epitheliochorial placenta, newborn piglets depend on immunoglobulins from the sow colostrum for protection against infections during the first weeks of life [19]. Sow vaccination is therefore a widely used strategy for the prevention of many porcine pathogens. However, recent reports indicate that the impact of MDAs may be more complex than previously perceived [8,20–25]. Described examples of "unwanted" effects of MDAs include impaired/delayed development of immunity, prolonged shedding periods, an increased risk of enzootic IAV infection at the herd level and vaccine associated enhanced respiratory disease (VAERD) [8,20,21,25–28].

The high mutation rate of RNA viruses enable them to rapidly evolve variants with a better fitness and/or modified antigenicity, capable of evading the immune system [29,30]. The surface protein hemagglutinin (HA) of IAV is more variable than the other viral proteins, consistent with the fact that it is the major target for neutralizing antibodies [31–34]. Especially mutations in the globular head of the hemagglutinin protein, which includes the receptor binding site for host cell entry and five specific antigenic sites/epitopes (Sa, Sb, Ca1, Ca2 and Cb) have been shown to modify the binding of neutralizing antibodies [35–39].

The continuous circulation of a huge variety of antigenically distinct variants of swIAV provides a significant challenge for the control of swIAV, because herd immunity may be compromised by introduction of new strains and/or by emergence of antigenically different variants within the herd. There is, however, a lack of controlled field studies on the swIAV dynamics in these herds.

A Danish sow herd that had been persistently infected with a swIAV of the H1avN1 subtype for years suddenly experienced an acute outbreak involving an H1avN2sw strain. By a co-incidence, this herd was included in another project and therefore we were able to perform a prospective study including observations and samplings both prior to, during, and after the outbreak. The aim of the study was to examine the clinical impact, the viral dynamics, as well as the genetic and antigenic variability of circulating strains prior to, during and after the acute outbreak. Following the acute outbreak, the herd decided to start a mass sow vaccination program, and therefore the study was extended to include exploration of the effects of a mass sow vaccination initiated during an acute outbreak.

## Materials and methods

### Ethical statement

The study was carried out in strict accordance with the guidelines of the Good Experimental Practices (GEP) standard adopted by the European Union, and all experimental procedures were performed in accordance with the recommendations provided by the National Veterinary Institute of Denmark. All samples were collected by a trained veterinarian and with the farmers consent. The Danish authorities do not require a specific license to obtain diagnostic samples in field settings according to the legislation LBK no. 474 of 15/05/2014.

### Herd

The herd was located in the northwestern part of Jutland, Denmark and had 600 sows with a main production of 30 kilos pigs and a small production of finisher pigs. The herd had a known health status according to the Danish Specific Pathogen Free program [40] termed "Blue SPF + AP2 + PRRS Type 1", indicating that the herd was serologically positive for *Actinobacillus pleuropneumoniae* type 2 and PRRSv type 1. However, both of these pathogens were under control. In addition, the health status specified that herd was declared free from *Mycoplasma hyopneumoniae*, *Brachyspira hyodysenteriae*, *Pasteurella multocida*, *Haematopinus suis* and *Sarcoptes scabiei* var. *suis*. The herd bought all new gilts from an external source, which had an identical health status. The replacement rate of the sows was approx. 50%/year. All piglets were weaned batch-wise at four weeks of age and placed in empty nursery stables where they were allocated to pens according to size. The herd had four farrowing stables and ten nursery stables. All stables were washed with high pressure and disinfected using hydrated lime between batches. No strict "all in all out" flow of pigs was maintained in any of the stables.

### Study design and sampling

As it was the plan to include the herd in another IAV study [41], it was screened for the presence of swIAV in December 2016. At the screening, 30 nasal swabs were collected which included five nasal swabs obtained from one-week-old piglets (farrowing unit), five nasal swabs of three-week-old piglets (farrowing unit), 10 nasal swabs from five-week-old weaners (nursery) and 10 nasal swabs from 9-week-old weaners (nursery).

In February 2017, the herd veterinarian reported an increase in respiratory problems in the farrowing unit, and increased secondary bacterial infections in nursery pigs. The 1st round of sampling was carried out from March to June 2017. The sampling round included four batches of sows with farrowing dates one-week apart. At farrowing, five piglets born from each sow were ear-tagged. Nasal swabs were collected from the ear-tagged pigs at week 1, 3, 5 and 10–12 and blood samples were collected at week 3 and at weeks 10–12. The exact same study design was conducted for the 2nd round of sampling, which was carried out from May to August 2017 after implementation of mass sow vaccination with Respiporc Flu®3 (IDT Biologika GmbH, Dessau-Rosslau, Germany) (Fig 1A and 1B). All sows with farrowing dates from May and onwards were vaccinated for the first time in the third week of March, and for the second time three weeks later. To avoid a mix of unvaccinated and vaccinated sows in the farrowing unit for the 2nd sampling, the first batch of sows included farrowed the last week of May, five weeks after their 2nd vaccination. The following sow batches were thereby six, seven and eight weeks post 2nd vaccination. All sows of the 1st and 2nd sampling round were randomly selected. A timeline showing the different sampling rounds in relation to IAV occurrence and vaccination is presented in Fig 1A.

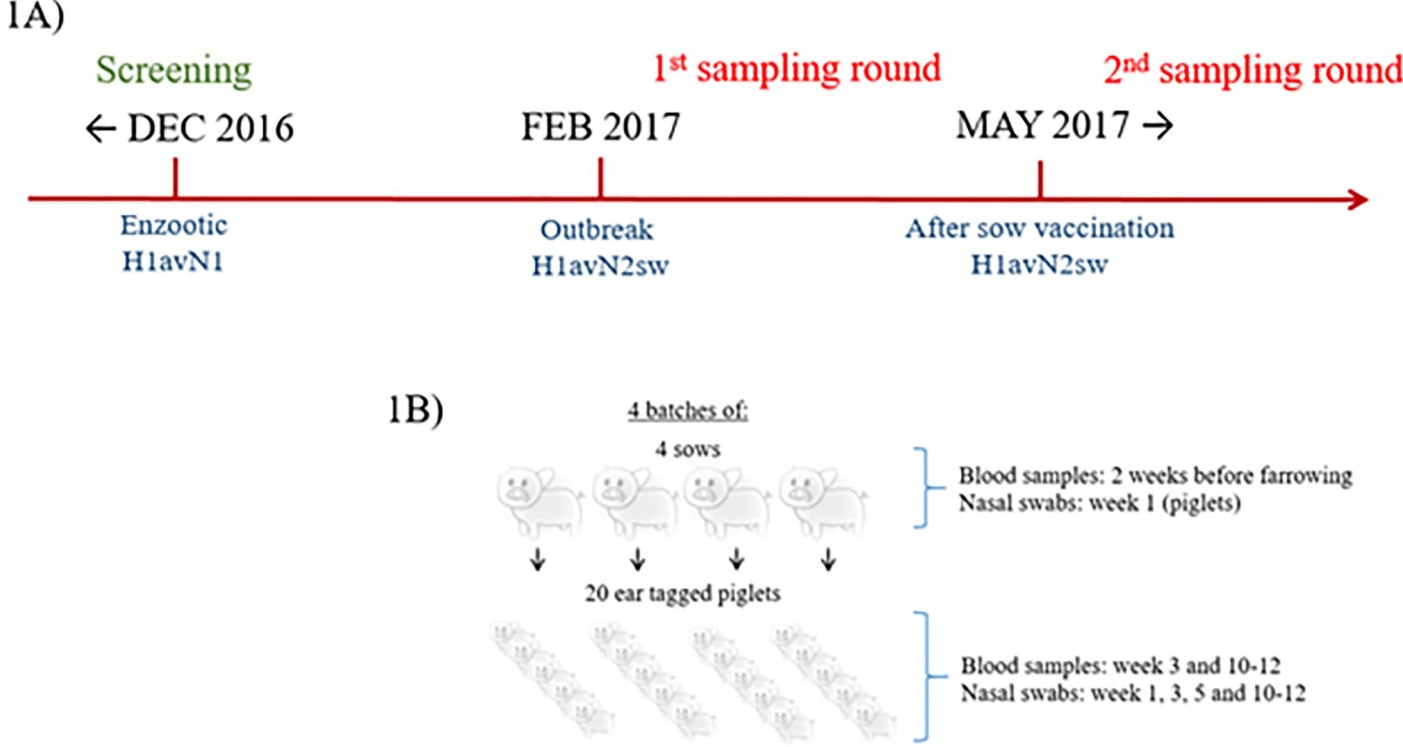

**Fig 1. Overview on the timeline of the study in relation to IAV occurrence and vaccination (1A) and the study design (1B).**

Blood samples and nasal swabs were collected from the sows and the piglets at different time points according to Fig 1B. Between two-five ml of blood were collected, using a vacutainer serum tube (Becton Dickinson, Denmark), from sows and piglets in *vena jugularis* and *vena cava cranialis* respectively. The blood samples were kept at 5°C for a maximum of 2 days. Subsequently, the samples were centrifuged at 3000rpm for 10 minutes, and the serum stored at -20 °C until test.

The nasal swabs were collected with a small or large rayon swab (Medical Wire, UK) according to the size of the animal. The swab was inserted and turned 360 degrees in both nostrils of each pig, and then immersed into the Sigma Virocult media (MWE, England). The samples were kept at 5° C for a maximum of 2 days until pooling and RNA extraction. Extracted RNA was kept at -80 °C until use.

### Clinical registrations

The clinical registrations were performed as previously described [41]. Briefly, a coughing index for the pen including minimum one ear tagged pig was calculated and individual clinical signs including dyspnea, lacrimation, nasal discharge, conjunctivitis, fecal soiling, body condition, limping and hernia were recorded for ear-tagged pigs.

### Pooling of nasal swabs, RNA extraction and quantitative real time RT-PCR

The pooling, RNA extraction and quantitative real time RT-PCR was performed as previously described [41]. Briefly, the nasal swabs were pooled litter-wise for the piglets and batch-wise for the sows. RNA was extracted from the pools using the RNeasy mini kit automated on the QIAcube (QIAGEN) using the large sample protocol version 2. For each extraction swIAV

positive and negative controls were included. Following extraction, five μL of RNA was used as a template in a previously published quantitative real time RT-PCR assay targeting the matrix gene influenza A [42], for determining if a pool was positive for IAV. Samples with a Ct value <36 was considered positive. If a pool was positive, the RNA was extracted from the individual samples and then tested by quantitative real time RT-PCR as described above. The positive individual samples with a Ct value <31 were tested in a previously described multiplex quantitative real time RT-PCR [41], to determine the IAV subtype. Samples with a Ct value <31 was considered positive in the given HA and NA assay included in the multiplex real time RT-PCR.

## Viral isolation and sequencing

At least one positive individual sample of each batch with a Ct value < 31 were chosen for PCR amplification of the HA and NA genes followed by Sanger sequencing as previously described [43]. In addition, the sample with the lowest Ct value from the initial screening and five samples with the lowest Ct value from the 1st and 2nd sampling round were chosen for isolation in Madin-Darby Canine Kidney (MDCK) cells and sequenced on the Illumina Miseq platform as previously described [43].

## Analyses of the viral sequences

The generation of consensus nucleotide and amino acid sequences of the Sanger's sequencing data and the Illumina sequence data was performed as previously described [43] using the program CLC Main Workbench version 8 for the Sanger sequencing reads and CLC Genomics Workbench version 11.0.1 for the Illumina reads. All sequences are available in NCBI Genbank with the following accession numbers: MN410726-MN410785 (Sanger sequences) and MN410796-MN410883 (Illumina sequences).

The consensus nucleotide- and amino acid-sequence of each gene (HA, NA, M, NP, NS, PB1, PB2 and PA) derived from the Sanger and Illumina sequencing were aligned using the MUSCLE algorithm [44] in CLC Main Workbench version 8. The subtype(s) of the IAV strain circulating in the herd were determined by constructing a phylogenetic tree (using neighbor joining) that included both contemporary HA and NA sequences, obtained in the Danish annual swine IAV surveillance, and also the HA and NA sequences from the present study, aligned using MUSCLE. Thereafter, the location of known antigenic sites (Sa, Sb, Ca1, Ca2 and Cb) of the H1 gene [35,37,38,45,46] were identified in the HA amino acid alignment of the present study, and examined manually for differences/mutations. For the remaining genes a BLAST analysis was performed against NCBI Genbank [47] to determine the closest sequence match and thereby the origin (avian or pandemic). Finally, the sequences were subjected to pairwise comparison, to reveal the overall sequence identity among sequences of the different samplings (screening and 1st and 2nd sampling rounds) and to the HA protein of the vaccine strain representing the lineage circulating in the herd.

**Viral evolution of the HA gene.** We used programs from the software package BEAST2 version 2.5.2 [48] to estimate the substitution rate for the HA gene both before vaccination (during the 1st sampling round) and after mass sow vaccination (during the 2nd sampling round). Specifically, the substitution model was specified to be HKY with gamma distributed rates over sites, with a strict clock model, and using tip dates (sampling dates). The following priors were specified: The tree model was set to "Birth Death Skyline Serial", which is used when lineages are sampled sequentially through time. The reproduction number was set to be

between 0 and 10 with a log normal distrubution. The "BecomeUninfectiousRate" was estimated to be approximately 52 per year (corresponding to an average time being infectious of 1 week) with a log normal distribution and $CI_{95\%} = [44.4–224]$. The clock rate was set as a log normal distribution with a mean value of 0.001 and, which is estimated to be substitution rate of RNA viruses, with a $CI_{95\%} = [3.95 \times 10^{-5}–0.005]$. The gamma shape prior and the kappa prior were left at the default values. A gamma distribution of the "origin prior" was chosen with an alpha value of 0.5 and a beta value of 2. Lastly, the sampling proportion prior was set to a log normal distribution with a mean value of 0.001 and $CI_{95\%} = [3.95 \times 10^{-5}–0.005]$. The chain length was set to 10.000.000 with a log every 1000, and the MCMC was run twice. The program BEAUti2 [48] was used to set up the analysis, and Tracer version 1.7.1 [49] was used to inspect the results and check for convergence of the MCMC runs.

The program CODEML in the program package PAML [50] was used to investigate whether there were positively selected sites in the two datasets. Specifically, we did this by comparing the fits of CODEML's Model 1a (M1a) and Model 2a (M2a) (NSsites = 1 and 2). In these models, selection is quantified using the dN/dS ratio (the ratio between the rate of non-silent substitutions per non-silent site and the rate of synonymous substitutions per synonymous site). A dN/dS ratio larger than 1 indicates the presence of positive selection (there are more amino-acid changing substitutions than expected for random reasons). M1a is a two-parameter model, which assumes two classes of codons, one class with negatively selected sites (dN/dS < 1) and one with neutral sites (dN/dS = 1), whereas M2a is a three-parameter model, which includes an additional class of positively selected sites (dN/dS > 1) [51]. If M2a fits the data significantly better than M1a (given the extra parameters in the model), then this is statistical evidence for the presence of positive selection in some codons. The Bayes Empirical Bayes (BEB) procedure [52] implemented in CODEML, was used to identify which sites that were positively selected. Model fits were compared using the Akaike Information Criterion (AIC), and Akaike weights, and also using likelihood ratio tests [53,54]. Moreover, an additional CODEML analysis, was used to determine the average global dN/dS (ω) value for the HA genes (NSsites = 0)[51,55].

We also used the program MrBayes [56] to estimate both clock rates and the presence of positively selected sites simultaneously. Specifically the codon model with gamma distributed rates was specified as: lset nucmodel = codon omegavar = ny98 rates = gamma, and report possel = yes site omega = yes. Node Dating was specified using the function "calibrate" to add a fixed sampling time to each sequence. The following priors were set for each data set: prset brlenspr = clock:uniform clockratepr = normal treeagepr = truncatednormal nodeagepr = calibrated. The data analysis was performed using two parallel runs for 3.000.000 generations with a sample frequency of 600. The phylogenetic tree was inferred in a Bayesian framework and with MCMC sampling of posterior probabilities. Tracer version 1.7.1 [49] was used to inspect results and check for convergence of the two MCMC runs. Tree visualization was performed using FigTree version 1.4.4 [57]

## Influenza ELISA

All blood samples were tested for antibodies against all Influenza A types using a commercially available blocking ELISA (IDEXX; Influenza A Ab Test; IDEXX Laboratories, Inc.). This test targets a conserved epitope in the nucleoprotein (NP) of influenza A virus. The OD values of the samples were divided by the OD value of the negative control to determine the S/N ratios. Samples were regarded as positive if they had an S/N ratio <0.6 and negative if it had an S/N ration ≥0.6.

## Hemagglutinin inhibition (HI)-test

Following analysis of the full genome sequences obtained from the viral isolates, one viral isolate from each sampling (screening, 1st sampling and 2nd sampling), were selected for the use in the HI-tests. The HI-tests were performed to determine the specific antibody titers of the sows included in the 1st and 2nd sampling rounds, against the three different viral strains of the study: the enzootic IAV strain found at the screening test "P5-U4" (A/sw/Denmark/P5U4/2016(H1N1): accession no. for HA; MN410806) and two different variants of the epizootic IAV strain–"HB4" (A/sw/Denmark/HB4280U1/2017(H1N2): accession no. for HA; MN410800) isolated before and "VB4" (A/sw/Denmark/VB4379U3/2017(H1N2): accession no. for HA; MN410805) isolated after the implementation of mass sow vaccination. To test if the vaccinated sows had indeed been vaccinated, an additional HI-test was performed including an H3N2 isolate, similar to the H3N2sw included in the vaccine. Immune sera raised against Respiporc FLU 3 and against the H1N1 component of the vaccine was used as controls. First, a hemagglutination (HA) test was performed to determine the HA titer of each viral isolate, and four HA-units (HAU) of the viral isolates were used as antigen for the HI-test. The sera were inactivated at 56˚C for 30 minutes and then treated with receptor-destroying enzyme (RDE). Then, the sera were mixed with 50% erythrocytes to remove specific inhibitors of haemagglutination and agglutination factors. Two-fold serum dilutions were tested against the four isolates, starting at a dilution of 1:20 followed by the addition of 0.6% guinea pig red blood cells. The titers were expressed as the reciprocal of the highest dilution of serum inhibiting the four HAU, and subjected to log2 transformation for statistical analysis. The average mean log2 values were subsequently converted back to average HI-titers. An HI-titer <20 were considered negative.

## Statistics

For each sampling round, a statistical analysis was performed comparing the prevalence of IAV-positive and IAV-negative individuals at each sampling time (weeks 1, 3, 5 and 10–12) with the presence of one of the clinical signs registered at the individual level using a Pearson's Chi squared Test. The same test was performed for comparing a difference in prevalence of seropositive and seronegative pigs at week 3 and week 10–12 between the two sampling rounds.

For an overall statistical comparison of means from the normally distributed data (CI, HI-titer, substitution rate and omega values etc.) a Student's t-Test was performed comparing both IAV positive and negative pigs and comparing the results of the 1st and 2nd sampling round [58].

All statistical analysis and graphs were completed using GraphPad Software [58] and Microsoft Excel. P-values below 0.05 were considered statistically significant.

## Results

The initial plan was to include the herd in another study describing the swIAV dynamics in enzootic infected herds, and therefore screening samples were obtained in December 2016. The collection of screening samples provided us with the enzootic H1avN1 strain for comparison to the epizootic H1N2sw strain, which was introduced in the herd around February 2017. In total, 30 screening samples were collected from two different age groups in the farrowing and nursery unit, respectively. At the 1st and 2nd sampling round 16 sows and 80 ear-tagged pigs were included, respectively. The number of ear-tagged pigs varied slightly between samplings due to mortality or ability to locate the ear-tagged pigs (Table 1).

**Table 1. Number of pigs testing positive for IAV in nasal swabs in the different batches at week 1, 3, 5 and 10–12 during the 1st and 2nd sampling.**

| | Batch 1 | | Batch 2 | | Batch 3 | | Batch 4 | | Total | |
|---|---|---|---|---|---|---|---|---|---|---|
| **Sampling:** | **1st** | **2nd** | **1st** | **2nd** | **1st** | **2nd** | **1st** | **2nd** | **1st** | **2nd** |
| Week 1 | 16/20 | 4/19 | 12/19 | 0/20 | 9/18 | 3/19 | 19/19 | 0/17 | 56/76 | 7/75 |
| Week 3 | 0/18 | 4/18 | 0/19 | 15/20 | 0/17 | 6/18 | 1/19 | 7/15 | 1/73 | 32/71 |
| Week 5 | 0/18 | 8/18 | 0/19 | 4/20 | 0/18 | 14/19 | 0/19 | 14/17 | 0/74 | 40/74 |
| Week 10–12 | 0/17 | 1/18 | 0/18 | 3/19 | 0/14 | 2/19 | 0/18 | 5/16 | 0/67 | 11/72 |

The values are given as the number of pigs testing positive of IAV in nasal swabs out of the total number of pigs sampled at the given sampling time.

### IAV at the screening test

At the screening test in December 2016, the three- and five-weeks old pigs tested positive for IAV in nasal swabs, while the one-week old piglets and nine-week old weaners were negative.

### IAV and IAV antibodies– 1st sampling round (before mass sow vaccination)

Test of serum for antibodies against IAV by ELISA revealed that all of the sows of the four batches were seropositive two weeks before farrowing (Fig 2A). In total, 81% of the three-week old piglets were seropositive, whereas the number of seropositive piglets decreased to 31% at week 10–12 (Fig 2A).

At week 1, 73.7% of all the piglets tested positive for IAV in nasal swabs (Fig 2A). One of these piglets also tested positive at week 3, but the remaining pigs were negative for IAV at weeks 3, 5 or 10–12 (Table 1). Detailed results on viral shedding and antibody status of each individual sow and ear-tagged pig are available in S1 table.

### IAV and IAV antibodies—2nd sampling round (after mass sow vaccination)

Test of serum for antibodies against IAV by ELISA revealed that all of the sows in the four batches were seropositive two weeks before farrowing and that 82.3% of the three-week old piglets were seropositive. Conversely, a significantly lower ($p = 0.006$) number of 10–12 week-old-pigs were seropositive in the 2nd sampling round compared the 1st sampling round (31%). After vaccination, the number of pigs that tested positive for IAV in nasal swabs at week 1 was significantly reduced ($p < 0.001$) as only 9.3% were positive (Fig 2B). However, compared to the 1st sampling round, a significant increased ($p < 0.001$) number of IAV positive pigs at the subsequent three samplings were observed (week 3, 5 and 10–12), since 45.1%, 54% and 15.3% of the pigs tested positive for IAV in nasal swabs at week 3, 5 and 10–12, respectively (Fig 2B and Table 1). Detailed results of viral shedding and antibody status of each individual sow and ear-tagged pig are available in S1 table.

### Differences in viral shedding between the 1st and 2nd sampling rounds

The comparisons of the total number of individual pigs that were infected at least once during the study period, the number of pigs that tested positive at two consecutive sampling times, which we defined as "prolonged shedders" and the average Ct value at the different sampling times are shown in Table 2. No statistical significant difference was observed in the total percentage of pigs being infected at least once during the study period between the 1st and 2nd sampling rounds. However, a statistical significant increased number of "prolonged shedders" ($p < 0.001$) was observed at the 2nd sampling round after the implementation of mass sow vaccination. Moreover, a marked significant difference ($p < 0.0001$) was identified in the average

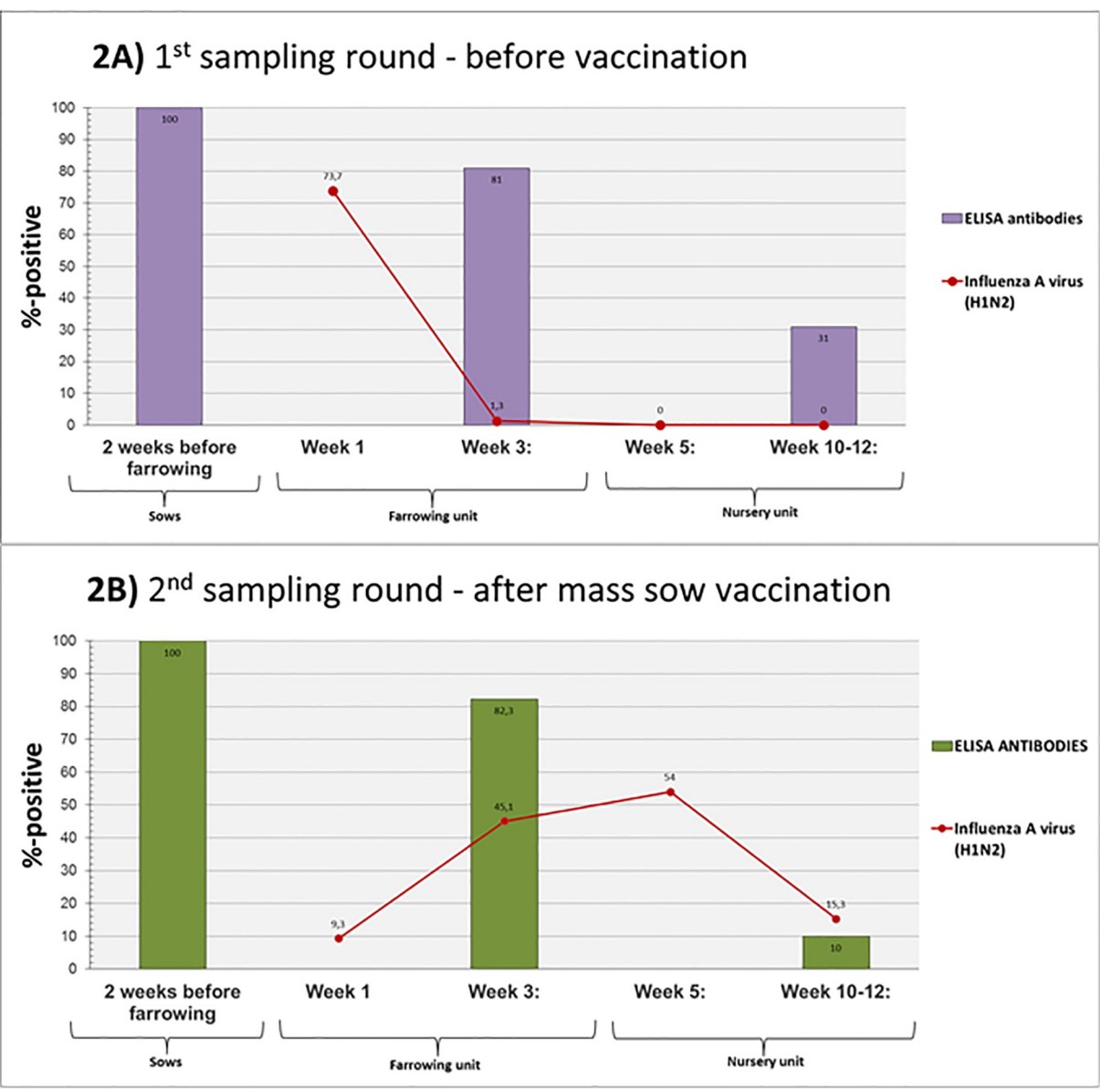

**Fig 2. The percentage of seropositive sows and pigs and the summed percentage of the number of pigs testing positive for IAV in nasal swabs at the 1st (2A) and 2nd (2B) sampling round.** The columns shows the percentage of seropositive sow and pigs. The blood samples were collected 2 weeks before farrowing from the sows and at week 3 and week 10–12 from the pigs born from the sampled sows. The red line show the summed percentage of pigs at each sampling time (week 1, 3, 5 and 10–12) testing positive for IAV in nasal swabs. "2A" presents the results of the 1st sampling round (before mass sow vaccination), and "2B" presents the results of the 2nd sampling (after mass sow vaccination).

Ct values between the two sampling rounds, indicating a significant decrease in viral shedding after implementation of the mass sow vaccination program.

## Clinical signs

The average coughing index (CI) for the IAV positive and negative litters/pens at the different sampling times and the presence of nasal discharge at the different sampling times compared to the number of pigs testing positive or negative for IAV are shown for the 1st and 2nd sampling rounds in Tables 3 and 4, respectively. No correlation was observed between the CI and

**Table 2. Prolonged shedders, total number of infected individuals and Ct values of the 1st and 2nd sampling.**

|  | 1st sampling | 2nd sampling | P-value |
|---|---|---|---|
| No. of prolonged shedders | 1.8% (1/56) | 28.3% (17/60) | <0.001 |
| No. of infected individuals | 73.7% (56/76) | 80% (60/75) | 0.467 |
| Average Ct value: |  |  |  |
| Week 1 | 16.7–34.2 (25.4) | 28.9–34.5 (31.3) | 0.0014 |
| Week 3 | 20.92* | 19.1–35.8 (30.7) | -* |
| Week 5 | - | 21.9–35.5 (31.9) | - |
| Week 10–12 | - | 23.7–34.4 (32) | - |
| Total: | 16.7–34.2 (25.11) | 19.1–35.8 (31.3) | <0.0001 |

The percentage of prolonged shedders is calculated based on the total number IAV positive pigs during the study.
The total percentage of infected pigs during the study is calculated compared to the number of pigs at the beginning
of the study. The range of Ct values obtained from the individual pigs is given for each sampling time, and the mean
Ct value for each sampling time is provided in the parentheses.
*In the 1st sampling round only one pig was positive at week 3 and therefore no range of Ct values are available and
no p-value for the differences between the 1st and 2nd sampling could be estimated.

the number of pigs with nasal discharge and the presence of IAV at the 1st sampling. Conversely, a significant correlation was seen between the presence of IAV and an increased coughing index at week 1, week 5 and over all samplings in the 2nd sampling round. Moreover, a significant correlation between the presence of IAV and nasal discharge was seen at week 1

**Table 3. Differences in mean coughing index between the 1st and 2nd sampling round.**

|  | Week 1 | | Week 3 | | Week 5 | | Week 10–12 | | Total | |
|---|---|---|---|---|---|---|---|---|---|---|
| *Sampling round*: | 1st | 2nd | 1st | 2nd | 1st | 2nd | 1st | 2nd | 1st | 2nd |
| Virus positive: | 0.14 | 0.13 | 1.14* | 0.37 | - | 0.23 | - | 0.02 | 0.2 | 0.23 |
| SD: | 0.1 | 0.11 | 0 | 0.19 |  | 0.2 |  | 0.01 |  | 0.21 |
| Virus negative: | 0.12 | 0.07 | 0.6 | 0.47 | 0.17 | 0.08 | 0.05 | 0.03 | 0.27 | 0.09 |
| SD | 0 | 0.06 | 0.38 | 0.4 |  | 0.1 |  | 0.03 |  | 0.18 |
| p-value | 0.77 | 0.06 | n/a | 0.50 | - | 0.05 | - | 0.51 | 0.49 | 0.0008 |

The p-value describes if a significant difference in the average coughing index was observed between virus positive and virus negative litters during the first and second
sampling respectively.
*only one registration as only one litter was positive and therefore no p-value has been included for this sampling.

**Table 4. Differences in the number of IAV positive and negative animals with nasal discharge between the 1st and 2nd sampling.**

|  | Week 1 | | Week 3 | | Week 5 | | Total* | |
|---|---|---|---|---|---|---|---|---|
| *Sampling round*: | 1st | 2nd | 1st | 2nd | 1st | 2nd | 1st | 2nd |
| **Virus positive:** | 60.7% (34/56) | 85.7% (6/7) | 100% (1/1) | 71.9% (23/32) | 0% (0/0) | 85% (34/40) | 61.4% (35/57) | 79.7% (63/79) |
| **Virus negative:** | 80% (16/20) | 39.7% (27/68) | 66% (48/72) | 74.4% (29/39) | 66% (49/74) | 67.5% (23/34) | 68% (113/166) | 56% (79/141) |
| **p-value** | 0.2 | 0.05 | 0.71 | 0.97 | - | 0.14 | 0.45 | 0.001 |

The parentages gives the number of pigs with nasal discharge out of the total number of positive or negative pigs.
*Week 10–12 were not included as nasal discharge was difficult to evaluate when using a nasal wire to restrain the pigs.

and over all samplings in the 2<sup>nd</sup> sampling round. No significant correlations between IAV and the presence of dyspnea, lacrimation, conjunctivitis, fecal soiling, body condition, limping and hernia were revealed in any of the sampling rounds.

### Subtyping and strain-characterization—Screening test (the enzootic IAV)

The results of the multiplex real time RT-PCR tests revealed that all the samples obtained at the time of the screening test were the H1avN1 subtype. This was consistent with the phylogenetic analyses of the HA and NA consensus sequences. The BLAST results revealed that all the internal genes were of avian-like H1Nx origin. The same subtype had been identified in the herd earlier trough diagnostic samples obtained by the herd veterinarian (personal communication).

### Subtyping and strain-characterization—1st sampling round (before mass sow vaccination)

The multiplex PCR tests revealed that all the samples obtained during the 1st sampling round, prior to mass-vaccination, belonged to the H1avN2sw subtype. This was in accordance with the HA and NA consensus sequences derived from the Illumina and Sanger's sequencing. The BLAST analysis of the internal genes revealed that the M, NP, PB1, PB2 and PA gene originated from the H1N1pdm09 subtype, whereas the NS gene were of avian-like H1Nx origin. A pairwise comparison of all HA sequences (n = 18) obtained from pigs sampled before vaccination revealed a close identity with 0–6 nucleotide differences corresponding to a sequence identity of 99.62–100%. The comparison also included two HA consensus sequences derived from the same pigs at two different sampling times (weeks 1 and 3), indicating that the pigs tested positive for the same strain for minimum two weeks. The full length of the HA protein was not obtained from all of the HA consensus sequences and therefore 21 nucleotides corresponding to seven amino acids were removed from the 5'end the all of the HA consensus sequences before further analysis. The HA proteins obtained from the 1st sampling round were aligned with the avian HA protein of the vaccine strain (Haselünne/IDT2617/2003, accession number: GQ161124), and the pairwise comparison revealed 91% amino acid identity, with several amino acid differences in antigenic sites (Fig 3).

### Subtyping and strain-characterization—2nd sampling round (after mass sow vaccination)

The multiplex PCR revealed that all the positive samples obtained from the 2<sup>nd</sup> sampling round were of the H1avN2sw subtype. This result was consistent with the HA and NA consensus sequences derived from the Illumina and Sanger's sequencing. Equal to the 1st sampling, the BLAST analysis of the internal genes revealed that the M, NP, PB1, PB2 and PA gene were of H1NXpdm09 origin, whereas the NS gene were of avian-like H1Nx origin. A pairwise comparison of all HA sequences (n = 19) from pigs sampled after vaccination revealed a close identity with 0–15 nucleotide differences corresponding to a sequence identity of 99–100%. The comparison also included seven HA consensus sequences derived from the same three pigs at two-three different sampling times, revealing that these pigs were positive for the same strain for two weeks and in some cases (pig 326 and 361) up to seven weeks (S1 table). To ensure an equal length of the HA sequences 21 nucleotides corresponding to seven amino acids were trimmed from the 5'end before further analysis.

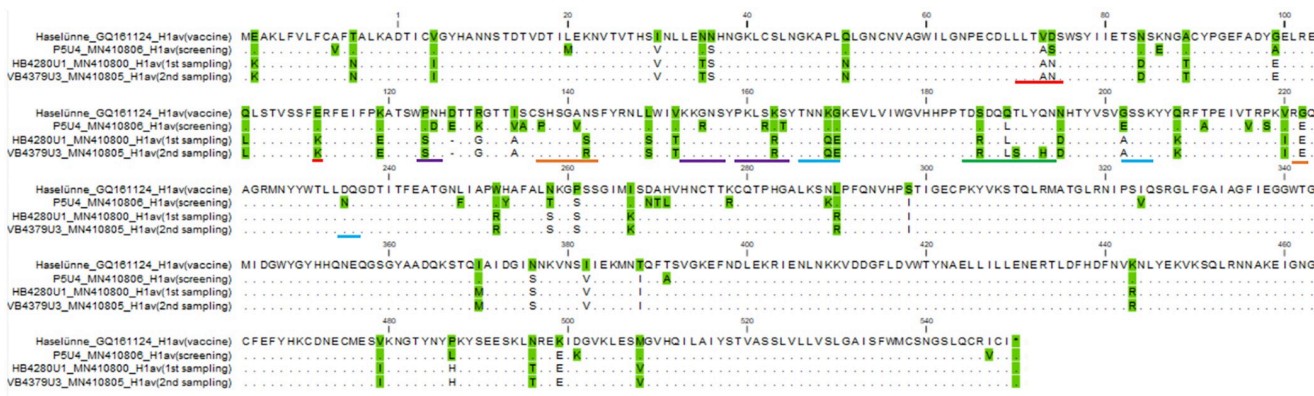

**Fig 3. Alignment of HA proteins obtained at screening, the 1ˢᵗ and 2ⁿᵈ sampling round and from the vaccine strain (Haselünne/IDT2617/2003 (H1N1)).** Dots indicate identical amino acids among the HA proteins, whereas a green background indicates differences between at least one of the four HA proteins. The colored underlines represent the five antigenic sites Sa (purple), Sb (green), Ca1 (blue), Ca2 (orange) and Cb (red).

## Comparison of HA consensus sequences obtained at the screening test and during the 1ˢᵗ sampling round

The avian-like HA genes from isolates collected at the screening test and at the 1ˢᵗ sampling round revealed a sequence identity of 87% at the nucleotide level and 89% at the amino acid level, which equaled 59–61 amino acid differences. The majority of the differences were present between amino acid Nos. 74–290 (numbering from DTIC), which includes the HA1 part of the HA gene, that encodes the globular head and contains the receptor binding site for host-cell-entry. Thirteen of the 54 differenes were present in specific antigenic sites (Sa, Sb, Cb, Ca1 and Ca2), including S74N, P124S, D125N, P137S, N142S/R, R155G, R162S, K163R, T164S, K169Q, G170E, S186R, N195D and E202A (numbering from DTIC). An alignment of HA proteins representing each sampling (screening and 1ˢᵗ and 2ⁿᵈ sampling round) can be visualized in Fig 3 along with the HA protein of the avian H1N1 vaccine strain (Haselünne/IDT2617/2003, accession number: GQ161124) covering the H1av component.

## Comparison of the HA consensus sequences obtained at the 1ˢᵗ and 2ⁿᵈ sampling rounds (before and after mass sow vaccination)

Only minor differences were observed between the consensus sequences obtained at the 1ˢᵗ and 2ⁿᵈ sampling rounds (before and after the implementation of mass sow vaccination). On the nucleotide level, the HA genes were between 99–100% identical on nucleotide level and 98–100% identical on the amino acid level, corresponding to 0–8 amino acid changes. The amino acid HA sequences of both the 1ˢᵗ and 2ⁿᵈ sampling round contained a deletion at position 127 and therefore amino acid numbers above 127 would have 1 added to them to correspond to the current H1 numbering, from the DTIC. Some of the amino acid differences between the sequences obtained from the 1ˢᵗ and 2ⁿᵈ sampling rounds, were in specific antigenic sites (Fig 3) and included S142R (Ca2), which were present in 8/19 sequences from the 2ⁿᵈ sampling, T190A/S/N (Sb) present in 6/19 sequences form the 2ⁿᵈ sampling round and Q193H (Sb) present in 1/19 sequences from the 2ⁿᵈ sampling round.

## Comparisons of NA and the internal genes obtained at the 1ˢᵗ and 2ⁿᵈ sampling rounds (before and after mass sow vaccination)

The NA gene sequences of the H1avN2sw strain before and after vaccination were between 99–100% identical corresponding to up to 14 nucleotide differences. Amino acid differences

between isolates from the 1st and 2nd sampling rounds were identified in ten positions, however, the majority of differences were only found in one or two sequences. The similarity of the remaining genes between the 1st and 2nd sampling rounds varied among genes but was generally high with a maximum of ten nucleotide differences. Only one amino acid difference in the PB1 gene (V724I) were consistent in all the sequences of the 2nd sampling round compared to the sequences of the isolates from the 1st sampling round. The number of nucleotide differences and the location of the amino acid differences in the NA- and internal genes are listed in S2 table.

## Viral evolution of the HA gene—1st sampling round (before mass sow vaccination)

Based on analysis using BEAST, the substitution rate for the HA gene before vaccination was estimated to be 0.00316 substitutions per site per year (SEM = 0.000032) corresponding to an average of 5.04 nucleotide substitutions per year for the entire gene (1596 nucleotides long in this dataset).

CODEML analysis of the 18 HA sequences derived from pigs of the 1st sampling round (before mass sow vaccination) suggested that positive selection was mostly absent. Thus, likelihood ratio testing indicated that M2a did not fit the data significantly better than M1a ($p > 0.05$). In addition, the Akaike weights for M1a and M2a were 0.63 and 0.37 respectively, suggesting somewhat higher support for the model without selection. Under model M2a (which was only weakly supported) there was some evidence for positive selection site at position 203 (probability of being positively selected, Pr+, estimated to be 0.81). The global dN/dS (ω) value was estimated to be 0.31, i.e., on average the gene is under medium strong negative selection (the tendency is to conserve the sequence).

The CODEML analysis makes no assumption about the substitution rate being clock-like (rates are instead free to vary on each branch of the phylogeny). However, since there is evidence that influenza sequences typically do evolve according to a molecular clock, it can be advantageous to use a model that explicitly makes that assumption (and which will then use much fewer parameters). We therefore also analyzed the data using MrBayes, and a model including both a strict clock rate, and a codon-based substitution model for estimating dN/dS rates (S1 Fig). In agreement with the CODEML analysis, there is weak evidence for positive selection on position186 (Pr+ = 0.54), with an estimated dN/dS = 1.5. The average dN/dS rates for the negatively and positively selected sites were estimated to be 0.29 (negatively selected) and 1.81 (positively selected). Table 5 list estimated dN/dS and Pr+, and also whether the codon is a known antigenic site, for the codons with most support for being positively selected.

## Viral evolution of the HA gene—2nd sampling (after mass sow vaccination)

Using BEAST, the substitution rate for the HA sequences after vaccination was estimated to be 0.00357 substitutions per site per year (SEM = 0.0000176), corresponding to 5.7 nucleotide substitutions per year for the entire HA gene. This is 12% higher than the rate estimated before mass sow vaccination (significantly different with $p < 0.001$).

CODEML-based analysis of the 19 HA sequences derived from pigs after vaccination strongly supported the presence of positive selection. Specifically, the Akaike weight for M1a and M2a was 0.00035 and 0.9996 respectively, and M2a (which includes a class of positively selected sites) thus has much higher support than M1a. Under model M2a, positions 159 and 207 showed very strong evidence (>99%) of being positively selected. An additional 9 sites were identified as being positively selected with a lower probability (<95%) (Table 5). In agreement with these results, the average dN/dS (global ω) value for the entire HA gene was

**Table 5. Identification of positively selected sites of the HA gene by MrBayes and CODEML.**

| 1st sampling round (before vaccination) | | | | 2nd sampling round (after vaccination) | | | |
|---|---|---|---|---|---|---|---|
| Amino acid change: | ω value Mb/CO | Pr+: Mb/Co | Antigenic site: | Amino acid change: | ω value Mb/CO | Pr+ Mb/CO | Antigenic site: |
| I5V | 1.1672 | 0.3905 | - | N-5S | 2.1589 | 0.4676 | - |
| S142R | 1.1689 | 0.3913 | Ca2 | K-2T | 3.2259/7.241 | 0.7100/0.821 | - |
| **R186S** | 1.4519/4.709 | 0.5427/0.806 | Sb | **A-1D** | 2.5500/4.835 | 0.5521/0.53 | - |
| A224E | 1.1772 | 0.3952 | - | **T2A** | 2.5533/4.99 | 0.5527/0.548 | - |
| T368P | 1.1782 | 0.3957 | - | D84N | 2.2352 | 0.4841 | -* |
| L407M | 1.1743 | 0.3938 | - | **S142R** | 4.1941/8.612 | 0.9715/0.999 | Ca2 |
| | | | | **R186S** | 3.0122/5.604 | 0.6562/0.631 | Sb |
| | | | | **T190N/S/A** | 4.1222/8.57 | 0.9431/0.992 | Sb |
| | | | | **Q193H** | 2.5254/5.47 | 0.5469/0.605 | Sb |
| | | | | D346N | 2.2431 | 0.4858 | - |
| | | | | **W348G** | 2.6210/5.643 | 0.5674/0.625 | - |
| | | | | **G360D** | 2.5860/4.778 | 0.5600/0.523 | - |
| | | | | **R403W** | 2.5194/5.023 | 0.5456/0.552 | -^ |
| | | | | **R454K** | 2.4940/4.98 | 0.5401/0.547 | -^ |

Numbering is based on the HA protein without the signal peptide, thereby initiating the numbering from the amino acids "DTIC". Due to the deletion at position 144, values above have been added one. Amino acid positions with a "-"indicates that the change occurred in the sequence prior to "DTIC". "ω value" gives the dN/dS ratio for the positive selected sites. "Pr+" gives the probability of the codon being positive selected. "Mb" gives the results of the MrBases analysis. "CO" gives the results of the CODEML analysis. Antigenic sites were defined as the previously published Sa, Sb, Ca1, Ca2 and Cb sites of H1 [35,36,38,45,46].The codon highlighted in bold are the codon positions, which were defined as positive selected sites in the both analysis.

* located in a B-cell epitope identified in H1N1pdm09 [59].

^ located in a T-cell epitope identified in human H1[60].

estimated to be 0.35, and thereby slightly higher than that of the HA sequences before vaccination (0.31).

Analysis using MrBayes (where the model includes both a constant, clock-like substitution rate and a codon-based substitution model for estimating dN/dS) supported the conclusions from the CODEML-based analysis (Table 5 and S2 Fig). Thus, several sites now have support for being positively selected, with substantially higher estimated dN/dS rates. Specifically, the estimated dN/dS rates were 0.05 for negatively selected sites and 4.23 for positively selected sites. The dN/dS value for positive selection was significantly increased (p-value <0.0001) compared to the dN/dS value identified among the sequences before vaccination. The positively selected sites are presented and compared to the CODEML analysis in Table 5. Six of the sites were located in previously known antigenic sites or in B-cell or T-cell epitopes, and included the two codons (159 and 207) which were identified through both analyses, as having the highest probability of being positively selected.

### Hemagglutinin inhibition test (HI-test)

The results of the HI-test of the sow sera from the unvaccinated sows during the 1st sampling and HI-titers of the vaccinated sows of the 2nd sampling are shown in Fig 4 and S1 table. The sera was tested against three different virus isolates: P5-U4, which were isolated from one of the screening samples, HB4 which was isolated from one of the nasal swabs from the 1st sampling, and VB4 isolated from the 2nd sampling round. The VB4 isolate had three amino acid changes (S159R, T207S and Q210H) compared to the HB4 isolate. For the HB4 isolate, serum dilutions were only made until 1:640 due to a lack of viral isolate.

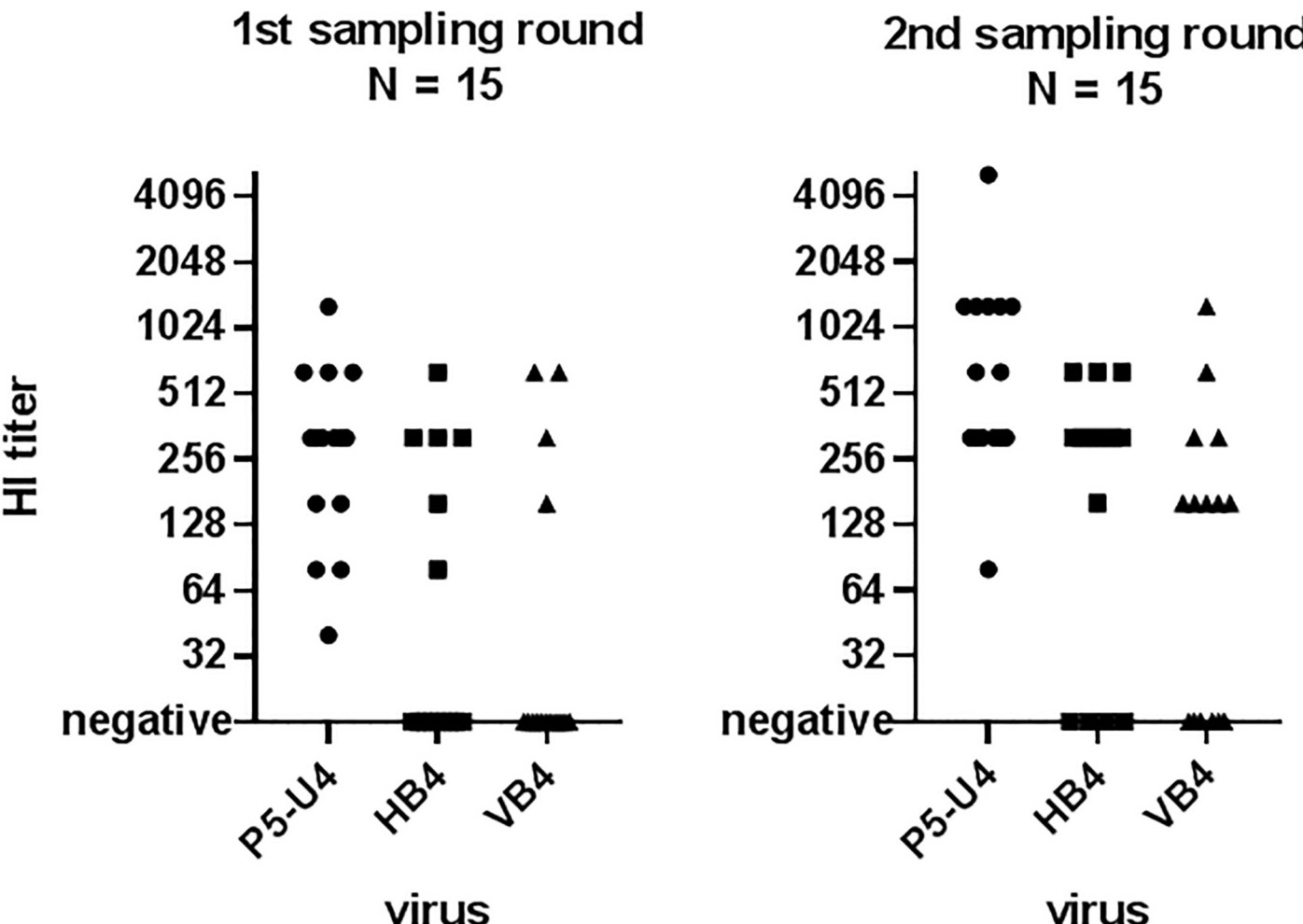

**Fig 4. Results of hemagglutination inhibition (HI) test of sow sera collected during the 1st (left) and 2nd (right) sampling round.** Each sow sera was tested against three different virus: P5-U4 (A/sw/Denmark/P5U4/2016(H1N1)) isolated from a pig sampled during the screening; HB4 (A/sw/Denmark/HB4280U1/2017(H1N2)) sampled from a pig during the 1st sampling round and before vaccination and VB4 (A/sw/Denmark/VB4379U3/2017(H1N2)) collected during the 2nd sampling round after start of mass sow vaccination. Negative samples were samples with a HI titer below 20.

The results revealed that all of the sows from the 1st sampling round had antibodies that reacted with the enzootic H1avN1sw strain (P5-U4) (mean log2 = 8.05; mean titer: 266). However, only six of the fifteen sows had antibodies towards the epizootic H1avN2sw strain (HB4) (mean log2 = 7.98; mean titer: 254) isolated before start of mass vaccination and, similarly, only four of the fifteen sows had antibodies towards the epizootic H1avN2sw strain (VB4) (mean log2 = 8.57; mean titer: 381) isolated after mass vaccination. All the vaccinated sows of the 2nd sampling round reacted with the enzootic H1avN1 (P5-U4) strain, and with a significant (p-value 0.025) higher titer (mean log2 = 9.26/mean titer: 612) than the sows of the 1st sampling round. Furthermore, an increase in the number of vaccinated sows (11/15) with antibodies reacting against the epizootic strain isolated prior to vaccination (HB2) was observed along with a higher, but not statistically significant, average titer of 364 (mean log2 = 8.5) (p = 0.209). Interestingly, only 9 out of 15 of the vaccinated sows reacted against the strain isolated after start of vaccination (VB4) albeit those that reacted had a high mean titer of 275 (mean log2 = 8.1). The sow sera of the vaccinated sows were also tested against an H3N2sw

strain with high level of genetic similarity to the vaccine strain. The results of this test revealed that all vaccinated sows reacted against the H3N2 and with very high titers >1280, indicating that the sows had indeed been vaccinated.

## Discussion

In this study, the effect of mass sow vaccination in relation to an outbreak with a new swIAV strain in a previously persistently infected herd was investigated. A unique dataset was collected before, during and after the outbreak. Samples collected during the acute outbreak revealed that the infections with IAV solely occurred in the farrowing unit, with the vast majority of piglets being infected at week 1. This infection pattern clearly changed during the 2nd sampling round, which was conducted after the implementation of mass sow vaccination. A clear delay in onset of infection was observed, as very few piglets were infected at week 1. However, at week 3, almost 50% of the piglets were positive for IAV in nasal swabs, resulting in a high number of IAV positive piglets at weaning. At weaning (week 4), the piglets were mixed into the nursery, providing new naïve individuals for infection, which was most likely the reason for the peak of infection observed at week 5. The presence of IAV circulation in the nursery unit resulted in the presence of IAV positive pigs at the end of the nursery period, which increased the risk of IAV being transferred into the finisher unit. It should be noted that as the RNA concentration was not measured prior to performing the real time RT-PCR, and there was a risk of pigs testing false negative for IAV. However, the marked difference in the infection pattern between the 1st and 2nd sampling rounds would still be evident. The delayed infection time and the significantly lower viral load observed in the pigs after mass sow vaccination, suggested that the vaccine provided some level of protection through the transfer of MDAs to the piglets. However, as previously described, the presence of MDA at the time of infection can increase the individual shedding time [8,21,61]. Indeed, a marked increase in numbers of prolonged shedders, after the use of mass sow vaccination, was observed in this study. This increase in prolonged shedders and the delay in infection time resulted in a higher number of IAV positive pigs being moved around the production system, consequently spreading the IAV to all age groups present in the herd. This observation supports the modeling performed by Cador et al. [20] who concluded that the presence of MDA would extend the IAV persistence within the herd. Another consequence of the presence of MDA at the time of IAV infection can be a suppressed active immune response, which has been described by several studies both in regards to neutralizing antibodies, IgG, IgA and the T-cell responses [21,25–27]. In this study, it was observed that significantly fewer pigs were seropositive at weeks 10–12 from the 2nd sampling (after mass sow vaccination) compared to the 1st sampling (before mass sow vaccination), suggesting that the presence of MDAs, also in this study, impaired the development of antibodies. The study did not provide data to elucidate if these seronegative pigs are protected against subsequent infection with the same IAV strain despite the lack of measurable antibodies or if they have developed a sufficient memory response. Re-infection with the same subtype after the MDA has declined has only been shown in one study [21] and needs to be investigated further. It should be noted that the MDAs present during the 2nd sampling round were probably stimulated both by the vaccine and by natural exposure to the epizootic strain. Thereby, the study mainly highlights the effects of having heterologous antibodies (1st sampling round) and homologous antibodies (2nd sampling round) towards the epizootic strain. Moreover, if the sows included in the second sampling had seroconverted towards the epizootic strain prior to vaccination, interference between pre-existing antibodies and the vaccine strain could be expected, possibly lowering the effect of the vaccine. However, mass sow vaccination in swIAV positive herds is extensively practiced in the field, and there is

a need for further studies investigating the effect of pre-existing immunity on swIAV vaccination efficacy.

The vaccine used in this study is approved for use in sows and pigs older than 56 days. The specific product characteristics (SPC) recommend a basic immunization of two doses applied with three weeks interval to obtain between four-six month immunity according to the age of the pig at the time of vaccination. However, when gestating sows are boosted two weeks before farrowing, the SPC of the vaccine claims to protect piglets against clinical signs of disease the first 33 days of life through transfer of MDAs [62]. In this study, mass sow vaccination was performed, meaning that while the sows included in the 2nd sampling all had received two vaccinations, they did not receive a booster two weeks before farrowing. However, this vaccination strategy is widely used in Denmark, and a protection of the piglets is expected. Clinical registrations were obtained, to reveal a possible clinical protection of the piglets, as a result of mass sow vaccination. Since different age groups became infected during the 1st (week 1) and 2nd (weeks 3–12) sampling rounds it is difficult to compare the results. However, a higher coughing index and the presence of nasal discharge was correlated with the presence of IAV in nasal swabs during the 2nd sampling round, indicating that vaccination of sows did not provide protection against upper respiratory tract infections. Since Denmark is almost free of swIAV of the H3NX subtypes, antibodies against this subtype can be used as a marker of vaccination with quite high sensitivity, and therefore we could confirm that the sows of the 2nd sampling round had indeed been vaccinated.

The results of this study indicated that mass sow vaccination during an acute outbreak might not be an optimal control strategy if the goal is to protect piglets against infection. However, it should be emphasized that the study was only performed in one herd. Furthermore, it is also important not to undermine the effect the vaccine might have had on protection of the sows, both during the gestation and when entering the farrowing unit, were circulation of IAV thrived [41]. In general, vaccination of naïve herds without IAV circulation is warranted, as an IAV introduction probably will lead to a major outbreak until the herd immunity has been built up. On the other hand, before initiating mass sow vaccination it is important to consider the impacts of the MDAs on the transmission dynamics.

Another important aspect with regard to protection achieved through antibodies was demonstrated in this study, as the herd, which was persistently infected with an H1avN1 strain, had a massive outbreak by an H1avN2sw strain. These two IAV strains have the same avian-like H1 gene, and thereby some level of cross protection was expected. However, it was clear from the 1st sampling round that most of the piglets became infected in week 1, despite most of them likely receiving MDAs from their seropositive mothers. Characterisation of the genetic differences of the enzootic strain found at the screening test and the epizootic strain found during the 1st sampling revealed major differences, suggesting that the outbreak strain was introduced into the herd from an external source. The influx of external gilts into the herd could be a possible route of novel swIAV introductions. The HA genes, while both being of avian-like H1 origin, were only approx. 88% identical, with several amino acid differences. A large proportion of these differences were located in parts of the HA gene encoding the globular head, and in locations corresponding to specific antigenic sites. Furthermore, the HI-results revealed that there was a weak cross-protection between the two strains, as 100% of the sows of the 1st sampling round showed a reaction against the enzootic strain H1avN1 as opposed to only 40% showing a reaction against the epizootic strain H1avN2sw. The results indicated that some swIAV strains, belonging to the avian-like H1 clade, have undergone antigenic drift to a degree that almost has abolished serological cross-reaction. The fact that the majority of the amino acid differences were located in the globular head of the HA protein, suggests that changes in these sites could increase viral survival by avoiding antibody binding. In addition,

differences in the other genes might also have an impact on the level of cross-protection between the two strains. This also raises the question if the avian-like H1 strain included in commercially available vaccines, provides cross-protection to all the different variants of the H1 avian-like subtypes. Indeed, a German study has investigated the genetic and antigenic diversity of the avian-like H1Nx viruses in several European countries, and documented an extensive diversity within this subtype, revealing an antigenic difference of up to ten antigenic units (AU) (personal communication Professor Tim Harder, FLI, Germany). In humans, the seasonal IAV vaccines are updated when the new strain differs four AU from the vaccine strain. Thus, this clearly suggests that a more regular update of strains in IAV vaccines intended for use in swine could be beneficial, but unfortunately, the European Medical Authority (EMA) does not allow for updates of veterinary IAV vaccines without going through a new licencing. The vaccine strains included in vaccine used in the herd of this study, contains IAV strains that are 15–19 years old, and from the alignment it is observed that several amino acid differences are observed between the vaccine strain and the herd strains, including differences in antigenic sites. Genetic differences between herd strain and vaccine strain might impact the vaccine efficacy. However, antigenic cartography or experiment animal trials are needed to investigate this further.

Differences in evolutionary dynamics were also observed between the viral sequences obtained from the 1st sampling round (before vaccination) and the 2nd sampling round (after mass sow vaccination). The molecular clock-based analysis, which takes sampling dates into account, revealed a significant increase in the nucleotide substitution rate in sequences obtained from pigs originating from vaccinated sows, compared to the ones originating from non-vaccinated sows. Furthermore, analysis using CODEML and MrBayes showed a substantial increase in positive selection (both the number of sites, and the strength of selection) after vaccination. Interestingly, several of the positively selected sites found in the sequences were located in the globular head of HA and in known antigenic sites. An isolate containing mutations in three sites (positions 159, 203, and 207) with strong support for being positively selected, was included in the HI-test. The results of the HI test showed that the number of sows from both the 1st and 2nd sampling that developed antibodies that recognized this isolate was decreased compared to the number of sows that reacted with the isolate without these three mutations. Though not significant, this suggested that the mutations had an impact on antibody binding. In conclusion, the evolutionary analysis clearly showed that an increase in the general substitution rate of the HA gene and positive selection in codons encoding antigenic sites occurred between the 1st and 2nd sampling round and that this had a negative impact on antibody binding. It is known that the global population immunity against human seasonal IAV strains leads to selection of escape mutants [35,38,63,64], but to our knowledge, this has not been described for IAV infection in swine under field conditions. One explanation for the expanded diversity and the positive selection of escape mutants could be that the increase in the infection time of the individual pig we observed after mass sow vaccination increased the likelihood of mutations through viral drift. In the present study, it is not possible to conclude if this change in viral diversity were due to the use of the vaccine or if it was driven by the immunity raised against the circulating field strain. Further studies are needed to explore the impact of herd immunity, vaccination and the generation of escape mutants in swine.

## Conclusion

The results of this study provided unique data on a case, where a previously persistently infected herd experienced an outbreak with a new subtype of IAV and it was thereafter decided

to start mass sow vaccination. The genetic analysis revealed that differences within the same IAV subtype can lead to a lack of cross-protection toward a similar strain and consequently to an acute outbreak. The presence of homologous MDAs in piglets during infection resulted in a lower viral load, but also in an increased shedding time and an impaired active immune response to infection. The increased shedding time along with the presence of maternal derived antibodies could be factors driving the positive selection of the HA gene, which in time might lead to the generation of escape mutants.

## Supporting information

**S1 Fig. Bayesian strict molecular clock tree of the HA sequences of the 1st sampling.** The x-axis represents time in years. Node labels represent posterior probabilities. The sequences are named as follows: HB indicates that the sequence was obtained in the first sampling round, and the following cipher gives the batch-number. The next three ciphers gives the ear tag number of the pig and "U1", "U3", "U5" and "U10" indicates the sampling time according to week 1, 3, 5 and 10–12. "HA" indicates that the sequences encode the hemagglutinin gene. The name of the sequences of the phylogenetic tree corresponds to the specific sequence ID "X" of the sequences uploaded in NCBI Genbank (A/sw/Denmark/X/2017(H1N2)).
(DOCX)

**S2 Fig. Bayesian strict molecular clock tree of the HA sequences of the 2nd sampling.** The x-axis represents time in years. Node labels represent posterior probabilities. The sequences are named as follows: VB indicates that the sequence was obtained in the second sampling round, and the following cipher gives the batch-number. The next three ciphers gives the ear tag number of the pig and "U1", "U3", "U5" and "U10" indicates the sampling time according to week 1, 3, 5 and 10–12. "HA" indicates that the sequences encode the hemagglutinin gene. The name of the sequences of the phylogenetic tree corresponds to the specific sequence ID "X" of the sequences uploaded in NCBI Genbank (A/sw/Denmark/X/2017(H1N2)).
(DOCX)

**S1 Table. Detailed table of the viral shedding and the antibody status (ELISA and HI-test results) of the sows and ear-tagged pigs at the different sampling times.** The table presents the four different batches of sows and their respective piglets at the different sampling times during the 1st and 2nd sampling round. The pigs of the 1st and 2nd sampling round were ear-tagged with numbers ranging from 200–282 and 300–380 respectively, whereas the sows were number from one-16. The green color indicates that the sow/pig tested positive in the antibody ELISA, whereas the red color indicates that the sow/pig tested negative in the antibody ELISA. "+IAV" indicates the nasal swab of the individual pigs or sows tested positive in the real time RT-PCR targeting the matrix gene of IAV. If a box is empty it either indicates that the ear tagged pig is dead or not sampled. The HI-test results of the sow sera is highlighted in yellow, and represents the HI-titers towards the three different swIAV strain found in the herd; P5-U4 (A/sw/Denmark/P5U4/2016(H1N1)), HB4 (A/sw/Denmark/HB4280U1/2017(H1N2)) and VB4 (A/sw/Denmark/VB4379U3/2017(H1N2)).
(DOCX)

**S2 Table. Nucleotide and amino acid differences among NA and the internal genes of the sequences derived from the pigs of the 1st and 2nd sampling.** The first columns describes the different genes. The second column describes the results of the pairwise comparison performed on the nucleotide consensus sequences. The third column describes the differences in amino acids according to the IUPAC codes. The forth column gives the position according to numbering from the first Methionine. The fifth column gives the number of sequences which

had the given mutation compared to total number of sequences obtained from the samplings; 1st = 1$^{st}$ sampling and 2nd = 2$^{nd}$ sampling.

(DOCX)

## Acknowledgments

The authors would like to thank the participating herd and the herd veterinarian for their cooperation and kind help during the study.

## Author Contributions

**Conceptualization:** Pia Ryt-Hansen, Anders Gorm Pedersen, Inge Larsen, Jesper Schak Krog, Charlotte Sonne Kristensen, Lars Erik Larsen.

**Data curation:** Pia Ryt-Hansen.

**Formal analysis:** Pia Ryt-Hansen.

**Investigation:** Pia Ryt-Hansen, Lars Erik Larsen.

**Methodology:** Pia Ryt-Hansen, Anders Gorm Pedersen, Lars Erik Larsen.

**Project administration:** Lars Erik Larsen.

**Software:** Anders Gorm Pedersen.

**Supervision:** Inge Larsen, Jesper Schak Krog, Charlotte Sonne Kristensen, Lars Erik Larsen.

**Validation:** Lars Erik Larsen.

**Visualization:** Pia Ryt-Hansen.

**Writing – original draft:** Pia Ryt-Hansen.

**Writing – review & editing:** Anders Gorm Pedersen, Inge Larsen, Jesper Schak Krog, Charlotte Sonne Kristensen, Lars Erik Larsen.

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
