## [Decision Letter · Decision Letter 0]

3 Oct 2019

PONE-D-19-25024

Acute Influenza A virus outbreak in an enzootic infected sow herd: Impact on viral dynamics, genetic and antigenic variability and effect of maternally derived antibodies and vaccination

PLOS ONE

Dear Mrs. Ryt-Hansen,

Thank you for submitting your manuscript to PLOS ONE. After careful consideration, we feel that it has merit but does not fully meet PLOS ONE’s publication criteria as it currently stands. Therefore, we invite you to submit a revised version of the manuscript that addresses the points raised during the review process.

This study describes the dynamics and variation of swine Influenza viruses (SIV) in a swineherd in Denmark before and after vaccination, and how maternally derived antibodies alter the viral diversity and immune response. Three age groups of pigs (piglets, weanlings and sows) were examined. Viruses were isolated from nasal swabs and viral genomes sequenced using Sanger and Illumina sequencing and compared to the circulating viruses. The study shows that immunization did select for key HA epitopes in the progeny viruses. Testing HA-specific antibody titers pre- and post-immunization revealed that pigs responded to both the pre-vaccination enzootic strain as well as the novel outbreak strain. Vaccination provided little benefit. They observed prolonged viral shedding in piglets. The reviewers indicate their concerns, but please particularly address:

The notion that the 2017-outbreak virus ‘evolved’ from the 2016 virus and why this is the case based on the number of differences in the HA1 amino acid sequence. Describe other evidence for this.As the vaccination appears to provide limited protection, as all the pigs are already seropositive please clarify the pre- vs. post-vaccination results (1st vs. 2nd sampling). Please address comments from all reviewers, and clarify responses to comments particularly to reviewer 2.

We would appreciate receiving your revised manuscript by Nov 17 2019 11:59PM. To enhance the reproducibility of your results, we recommend that if applicable you deposit your laboratory protocols in protocols.io, where a protocol can be assigned its own identifier (DOI) such that it can be cited independently in the future. For instructions see: http://journals.plos.org/plosone/s/submission-guidelines#loc-laboratory-protocols

We look forward to receiving your revised manuscript.

Kind regards,

Ralph A. Tripp

Academic Editor

PLOS ONE

Journal Requirements:

1. In your Methods, please state the volume of the blood samples collected for use in your study.

2. Thank you for including your ethics statement:  "The study was carried out in strict accordance with the guidelines of the Good Experimental Practices (GEP) standard adopted by the European Union, and all experimental procedures were performed in accordance with the recommendations provided by the National Veterinary Institute of Denmark. All samples were collected by a trained veterinarian and with the farmers consent."

Please amend your current ethics statement to confirm that your named ethics committee specifically approved this study.

For additional information about PLOS ONE submissions requirements for animal ethics, please refer to http://journals.plos.org/plosone/s/submission-guidelines#loc-animal-research  

3. We note that you are reporting an analysis of a microarray, next-generation sequencing, or deep sequencing data set. PLOS requires that authors comply with field-specific standards for preparation, recording, and deposition of data in repositories appropriate to their field. Please upload these data to a stable, public repository (such as ArrayExpress, Gene Expression Omnibus (GEO), DNA Data Bank of Japan (DDBJ), NCBI GenBank, NCBI Sequence Read Archive, or EMBL Nucleotide Sequence Database (ENA)). In your revised cover letter, please provide the relevant accession numbers that may be used to access these data. For a full list of recommended repositories, see http://journals.plos.org/plosone/s/data-availability#loc-omics or http://journals.plos.org/plosone/s/data-availability#loc-sequencing.

Additional Editor Comments (if provided):

This study describes the dynamics and variation of swine Influenza viruses (SIV) in a swineherd in Denmark before and after vaccination, and how maternally derived antibodies alter the viral diversity and immune response. Three age groups of pigs (piglets, weanlings and sows) were examined. Viruses were isolated from nasal swabs and viral genomes sequenced using Sanger and Illumina sequencing and compared to the circulating viruses. The study shows that immunization did select for key HA epitopes in the progeny viruses. Testing HA-specific antibody titers pre- and post-immunization revealed that pigs responded to both the pre-vaccination enzootic strain as well as the novel outbreak strain. Vaccination provided little benefit. They observed prolonged viral shedding in piglets. The reviewers indicate their concerns, but please particularly address:

• The notion that the 2017-outbreak virus ‘evolved’ from the 2016 virus and why this is the case based on the number of differences in the HA1 amino acid sequence. Describe other evidence for this.

• As the vaccination appears to provide limited protection, as all the pigs are already seropositive, please clarify the pre- vs. post-vaccination results (1st vs. 2nd sampling).

• Please address general comments from all reviewers, and clarify responses to comments, particularly to reviewer 2.

Reviewers' comments:

Reviewer's Responses to Questions

**Comments to the Author**

1. Is the manuscript technically sound, and do the data support the conclusions?

Reviewer #1: No

Reviewer #2: Yes

2. Has the statistical analysis been performed appropriately and rigorously? 

Reviewer #1: No

Reviewer #2: Yes

3. Have the authors made all data underlying the findings in their manuscript fully available?

Reviewer #1: No

Reviewer #2: Yes

4. Is the manuscript presented in an intelligible fashion and written in standard English?

Reviewer #1: Yes

Reviewer #2: Yes

5. Review Comments to the Author

Reviewer #1: Summary-

The ability to monitor “real world” swine influenza outbreaks within large-scale production systems can provide invaluable information. Elucidating the impact and timing of maternal antibodies in accordance with vaccination in an ongoing outbreak in pigs can offer insight into future vaccine mitigation strategies. Taking into consideration that these types of studies are difficult to recapitulate due to cost, complexity, and happening in “real time” in a BSL-2 animal facility, there are some fundamental issues that the authors need to resolve so the reviewer and ultimately the reader can interpret the data appropriately. As a reviewer I’m left awfully confused as to the nature of the virus that was detected initially at the screening process and how that relates to the virus detected a few months later during the outbreak that was circulating before/after vaccination. As it appears that the herd was persistently infected with H1avN1 strain, had a massive outbreak of H1avN2sw in Feb. 2017 (likely from a 2nd different clade that didn’t evolve from the first in in late-2016)- to what extent vaccination would help afterwards with the pigs all being sero-positive from infection before vaccination (Fig. 2A) adds to the confusion as it relates to maternal antibody protection. Furthermore before vaccination all your sows have antibodies to the 2016 virus (Fig.3, P5-U4) while half seem to have antbodies to the 2017 virus whereas half do not have antibodies (Fig. 3, HB4) this difference is not parsed out in the data. Lastly, the data tables and figures is presented in a summary form and therefore makes it hard as to how this provides insight to other researchers as to best practices in the farm-setting. Regrettably the title of the paper suggests that there this manuscript provides advances in the dynamics, genetic/antigenic variability, maternal antibodies and vaccination- but you have not teased apart these processes but simply provided a summary of a complex outbreak on a farm with 2 antigenically distinct HA1av strains in which you tried to vaccinate to mitigate the impact that was minimally impactful due to pre-existing immunity already present. Most importantly the impression is given that the 2017 virus in the outbreak “evolved” from the 2016 virus and that simply is not the case based on the number of differences in the HA1 amino acid sequence.

General comments are noted in the hopes to improve the manuscript so that a reviewer can provide a better critique of the manuscript.

General Comments-

1. The purpose of describing the screening process is unclear. Line 239, what is the purpose of the 30 screening samples and what is the result? You mention the screening process in the study design on lines 98-100, but is this simply to obtain a 2016 virus for comparison? Please clarify.

2. It appears that all sows were sero-positive by ELISA (Fig. 2A). It appears all sows are sero-positive to the 2016 strain (Fig. 3, left) while half are to the 2017 strain and half are not (Fig, 3 left). How does vaccinating an already flu sero-positive make a difference? It seems vaccination provides no benefit.

3. As shown in Figure 1B, do you know which sow gave birth to each ear-tagged piglet? This is imperative in deciphering maternal antibody impacts. Showing the piglet data based on the mother sow serology data (Fig 3), and not simply barn location is important.

4. Figure 2 in connection with Table 1, is awfully hard to decipher, and just showing %positive does not do the data justice. Could they have been exposed to the virus a week or two earlier and were now recovered and that is why they were not detected in nasal swabs at week 1? These and other questions are not answered, we are left guessing with a simple %summary of the data. Because of this it makes the data minimally impactful, these pigs have different immune-exposure histories, it is not appropriate to lump the data like this.

5. Table 3- coughing index, this is rarely shown in the swine influenza literature as a measure of infection as pigs can be asymptomatic to an active infection depending on the strain. Some pigs don’t cough at all during infection. Since you only have 1 of 73 pigs actively shedding virus after week one in the first sampling and you found no real differences this does not justify a stand-alone table. This table can be summarized in one sentence instead that “while there was nasal discharge indicative of infection (Table 1), there were no discernable differences in coughing.” Alternatively this data should be shown in bar graph form for one to even interpret.

6. This ties into #1-2. For instance, you show in figure 2A that all the pigs in the first round have ELISA antibodies. How did they become sero-positive and to what extent? From the 2016 or 2017 outbreak? This isn’t really answered until Fig. 3. Also we have no idea if maternal antibodies for the piglets are derived from an infection or vaccination. It appears that vaccination was minimally effective since all your pigs were sero-positive to begin with so this make interpreting and vaccination data in 1st sampling round vs. 2nd sampling round difficult.

7. The impression is given (lines 340-348) that the outbreak virus H1 in 2017 somehow evolved from the circulating strain a few months earlier in 2016 because you use the word “mutation”. This is not known, as you very well could have had two co-circulating viruses on the farm and then through reassortment they share the other 7 segments besides the HA. Also in Table 5 data I find it very difficult to believe that immune pressure lead by vaccination lead to this many changes in the HA. Please provide evidence that this many HA1 can change due to vaccination pressure in this short of a time frame. A more plausible explanation is you have 2 co-circulating strains on the farm and the vaccine preferentially prevented on type while the other was allowed to persist. While I am not an evolutionary biologist I very rarely see dN/dS ratios presented in flu studies in which viruses are circulating in less than a year’s duration, these types of analysis are done to look at evolution over decades.

8. You state the amino acid numbering scheme is based on the first methionine. In the flu literature H1 numbering occurs after the polypeptide sequence is cleaved, are you using this numbering scheme?

9. The data would be highly strengthened if we were shown an amino acid alignment of the 2016 strain, the 2017 strain and the viruses sequences with difference in amino acids in the H1, if there are duplicate sequences they can be lumped, however Table 5 does not suffice for depicting the evolution. There isn’t even the known amino acid changes shown in “codon” in table 5.

10. The strains used in the vaccine are known, please state them and speak to them and if the amino acid sequence is known, maybe even include it in the alignment (suggested in #8).

11. All relevant viral segments for at least one represenative strain circulating on the farm in 2016 and in 2017 needs to be deposited into a proper publicly accessible database (NCBI) and referenced in the manuscript (A/Sw/xxx/2016 for example), this doesn’t need to be done with every isolate but virologists need a reference and you need to speak to that in alignments and phylogenetic trees to delineate the 2016 vs, 2017 outbreak.

12. In line with #10, the supplementary phylogenetic trees are suboptimal and provide minimal information as to the H1av diversity circulating in the country and how they related to the rest of Europe. You do not provide any real background information of the H1av subtype in the introduction and discussion. You mention your sequences on line 152 but what do they correspond to?

13. Line 53, source #17, I did not find any data in this source that clarified that this vaccine provides protection to the Denmark subtype. While this is inferred from the H1avN1 component in the vaccine provides protection against your H1av do you have a proper source for this subtype?

14. Line 58- another unwanted effect reported by others in the swine influenza literature is vaccine associated enhanced respiratory disease (VAERD), please consider citing.

15. Line 24 abstract- state what the subtype is and provide a reference strain (see #10).

16. Figure 3 would be better resprented if a supplementary table showed the HI titers for each of the 20 pigs.

17. Figure 2 needs more descriptive labels to reflect before and after vaccination.

18. Figure 3, as already mentioned, many of your sows are sero-postive to flu, this data indicated that vaccination really didn’t do much. Also, did you sequence the P5-U4, HB4, and VB4 viruses before doing the HI? Were they grown in MDCK cells? Please include in materials and methods.

19. Fig. 3 is limiting as we have no idea about the antigenic cross-reactivity of sera to these H1av viruses. Any naiive serum as a control? Any positive swine serum you could obtain specific to the vaccine component from another group or previous study? it appears that all pigs have immunity to the 2016 strain (1st round- P5-U4), approximately half have immunity to the 2017 strain (HB-4) this highly impacts how the piglets might respond with these maternal antibodies.

Reviewer #2: In this study, the authors describe the dynamics and variation of swine Influenza viruses (swIAV)in one swine herd in Denmark before and after vaccination and how maternally derived antibodies alter the viral diversity and immune response. The authors were able to sample three age groups of pigs, piglets, weanlings and sows when the herd was infected with an outbreak strain that replaced the enzootic strain and then after mass administration of a commercially available swine influenza vaccine. Viruses were isolated from nasal swabs from these animals; viral genomes were sequenced using Sanger and Illumina sequencing and compared to the circulating viruses. The authors discover that immunization did positively select for some key epitopes in the HA antigen in the progeny viruses that likely helped these viruses adapt. Testing HA specific antibody titers in these animals pre- and post-immunization revealed that animals responded to both the pre-vaccination enzootic strain as well as the novel outbreak strain; they also observed prolonged viral shedding in the piglets post vaccination sows and ascribe it to the presence of maternal derived antibodies.

The manuscript is well written and most data analyses are well thought out. Sequence data are submitted to NCBI and will be available upon publication.

The major critiques of the manuscript are as follows

1. Sample positivity is defined by qPCR Ct values <36. This is dependent on the amount of RNA utilized for the one-step qRT-PCR they employ in this manuscript. There is no mention of the amount of RNA from each sample that was used for the qRT-PCR assay. Please clarify that samples that were considered flu negative were not due to lack of sufficient RNA in the reaction. Please clearly state the amounts of RNA used for qRT-PCR.

2. Please show the multiple alignments of the original enzootic H1N1, HB4 and VB4 strain HAs as supplemental material.

3. Line 337. please clarify what two-seven weeks (is it twenty seven, is it two through seven).

4. In Table2, please provide range of the Ct values instead of the mean. This would help understand the spread of the data across each time point as well as compare run efficiencies.

5. What Ct threshholds were utilized in the multiplexing assays used (line 309)?

6. Please speculate on why a majority of the amino acid changes in HA were located in the positions 91-307 between screening test and 1st sampling isolates. Do these changes improve viral fitness?

6. PLOS authors have the option to publish the peer review history of their article (what does this mean?). If published, this will include your full peer review and any attached files.

Reviewer #1: No

Reviewer #2: No

---

## [Author Response · Author response to Decision Letter 0]

18 Oct 2019

Dear Editor and reviewers,

Thank you a lot for your comprehensive review and very useful comments that indeed will improve the paper. We have tried to address and if needed, to incorporate all of your comments as described in details below: 

Editor:

The notion that the 2017-outbreak virus ‘evolved’ from the 2016 virus and why this is the case based on the number of differences in the HA1 amino acid sequence. Describe other evidence for this.

- The 2017 virus did not evolve from the 2016 and I have now corrected the section which describes the differences between the 2016 strain and the 2017 strain, so it now does not contain the word “mutation”, which could indicate that the 2017 evolved from the 2016 strain (line 382). Furthermore, this has also been emphasized in the discussion, where it is now mentioned that the 2017-outbreak virus likely were introduced from an external source (line 568-570). 

As the vaccination appears to provide limited protection, as all the pigs are already seropositive please clarify the pre- vs. post-vaccination results (1st vs. 2nd sampling).

- The impact of the pre-existing immunity is now discussed (line 529-537). However, it is also underlined that in the majority of herds choosing to implement swIAV vaccination, there will be pre-existing immunity toward swIAV, which can impact the vaccine efficacy, which also highlight the importance of sharing the results of this study. 

In your Methods, please state the volume of the blood samples collected for use in your study.

- This has now been stated in line 127.

Please amend your current ethics statement to confirm that your named ethics committee specifically approved this study.

- The statement has now been edited and refers to the Danish legislation, which does not require a specific license for the samplings performed in this field study.

We note that you are reporting an analysis of a microarray, next-generation sequencing, or deep sequencing data set. PLOS requires that authors comply with field-specific standards for preparation, recording, and deposition of data in repositories appropriate to their field. Please upload these data to a stable, public repository (such as ArrayExpress, Gene Expression Omnibus (GEO), DNA Data Bank of Japan (DDBJ), NCBI GenBank, NCBI Sequence Read Archive, or EMBL Nucleotide Sequence Database (ENA)). In your revised cover letter, please provide the relevant accession numbers that may be used to access these data. For a full list of recommended repositories, see http://journals.plos.org/plosone/s/data-availability#loc-omics or http://journals.plos.org/plosone/s/data-availability#loc-sequencing.

- All sequences have been uploaded in NCBI Genbank and will be available upon publication. This statement and the accession number are now both included in the manuscript and in the revised cover letter, and the reference strains highlighted in the manuscript are presented with their specific accession number. 

We note that you have included the phrase “data not shown” in your manuscript. Unfortunately, this does not meet our data sharing requirements. PLOS does not permit references to inaccessible data. We require that authors provide all relevant data within the paper, Supporting Information files, or in an acceptable, public repository. Please add a citation to support this phrase or upload the data that corresponds with these findings to a stable repository (such as Figshare or Dryad) and provide and URLs, DOIs, or accession numbers that may be used to access these data. Or, if the data are not a core part of the research being presented in your study, we ask that you remove the phrase that refers to these data.

- As the data in the two cases are not core parts of the study the two phrases has been deleted from the manuscript. 

Reviewer #1: 

General Comments:

1. The purpose of describing the screening process is unclear. Line 239, what is the purpose of the 30 screening samples and what is the result? You mention the screening process in the study design on lines 98-100, but is this simply to obtain a 2016 virus for comparison? Please clarify.

- The purpose for the screening samples has now been emphasized in line 256-259.

2. It appears that all sows were sero-positive by ELISA (Fig. 2A). It appears all sows are sero-positive to the 2016 strain (Fig. 3, left) while half are to the 2017 strain and half are not (Fig, 3 left). How does vaccinating an already flu sero-positive make a difference? It seems vaccination provides no benefit.

- Thanks for pointing this out. We have added a paragraph in the discussion section where this is discussed (line 529-537). Moreover, it was not the same sows included in sampling round 1, which were later vaccinated and included in sampling round 2. This has now been underlined in the study design (line 121-122). 

3. As shown in Figure 1B, do you know which sow gave birth to each ear-tagged piglet? This is imperative in deciphering maternal antibody impacts. Showing the piglet data based on the mother sow serology data (Fig 3), and not simply barn location is important.

- Yes, as mentioned in the materials and methods the sows were sampled and identified before birth, and all the piglets were ear-tagged at birth. However, this has been further emphasized in line 113 and in the Figure text of Figure 2 (line 282). Moreover, an additional supplementary table (S1 table) has now been included in the manuscript, presenting individual sow and piglet data.

4. Figure 2 in connection with Table 1, is awfully hard to decipher, and just showing %positive does not do the data justice. Could they have been exposed to the virus a week or two earlier and were now recovered and that is why they were not detected in nasal swabs at week 1? These and other questions are not answered, we are left guessing with a simple %summary of the data. Because of this it makes the data minimally impactful, these pigs have different immune-exposure histories, it is not appropriate to lump the data like this.

- Figure 2 is meant to give an overview on the different infection patterns observed during the first and second sampling round, as clear differences in infection time is observed. However, the pigs were not sampled every week, so of course we could have missed some infections. However, it is still clear from the data that fewer pigs were infected at week 1, and as pigs shed IAV for approx. 1 week and there is an incubation period before they start shedding, it is very unlikely that the piglets has already recovered from an swIAV infection at one week of age. However, to allow the reader to access the data on individual pigs, another supplementary table (S1 table) has been added where the details of each litter and sow can be found in regards to both infection status, ELISA antibodies and HI-titers. 

5. Table 3- coughing index, this is rarely shown in the swine influenza literature as a measure of infection as pigs can be asymptomatic to an active infection depending on the strain. Some pigs don’t cough at all during infection. Since you only have 1 of 73 pigs actively shedding virus after week one in the first sampling and you found no real differences this does not justify a stand-alone table. This table can be summarized in one sentence instead that “while there was nasal discharge indicative of infection (Table 1), there were no discernable differences in coughing.” Alternatively this data should be shown in bar graph form for one to even interpret.

- You are right that in some herds/batches the swIAV infection may be subclinical, but we have previously shown that the coughing index in some cases are significantly related to infection with swIAV in pigs (Ryt-Hansen et al. Vet Res (2019)). Even though no difference in coughing index was seen during the first sampling round, there was an association between an increased coughing index during the second sampling, and the table thereby highlight that no clinical effect of the vaccination was observed. The caption now specifies that a p value was not calculated for week 3 during the 1st sampling round, as only one litter was swIAV positive. 

6. This ties into #1-2. For instance, you show in figure 2A that all the pigs in the first round have ELISA antibodies. How did they become sero-positive and to what extent? From the 2016 or 2017 outbreak? This isn’t really answered until Fig. 3. Also we have no idea if maternal antibodies for the piglets are derived from an infection or vaccination. It appears that vaccination was minimally effective since all your pigs were sero-positive to begin with so this make interpreting and vaccination data in 1st sampling round vs. 2nd sampling round difficult.

- As also emphasized in the answer to comment #4 a new section in the discussion has now been added to emphasize that the effects seen after vaccination both can be due to antibodies stimulated naturally in the sow or as a response to the vaccination.

7. The impression is given (lines 340-348) that the outbreak virus H1 in 2017 somehow evolved from the circulating strain a few months earlier in 2016 because you use the word “mutation”. This is not known, as you very well could have had two co-circulating viruses on the farm and then through reassortment they share the other 7 segments besides the HA. Also in Table 5 data I find it very difficult to believe that immune pressure lead by vaccination lead to this many changes in the HA. Please provide evidence that this many HA1 can change due to vaccination pressure in this short of a time frame. A more plausible explanation is you have 2 co-circulating strains on the farm and the vaccine preferentially prevented on type while the other was allowed to persist. While I am not an evolutionary biologist I very rarely see dN/dS ratios presented in flu studies in which viruses are circulating in less than a year’s duration, these types of analysis are done to look at evolution over decades.

- The word “mutation” has now been changed to “differences”, so that the reader does not interpret that the 2017 outbreak strain evolved from the 2016 strain. The 2016 and the 2017 strain were not detected at the same time point and there is no evidence that the two strains circulated together for a longer time period, but rather that the outbreak strain replaced the 2016 endemic strain. Furthermore, it has been emphasized in the discussion that the 2017 was most likely introduced into the herd from an external source (line 568-570)

- In the discussion it is clearly stated that we do not know if the evolution is driven by immunity stimulated by the vaccine or by natural immunity towards the herd strain, and that further studies are needed to investigate this. However, the evolutionary analysis clearly showed an increased selection pressure directed towards antigenic important sites in sampling round 2 after start of vaccination. 

- The dN/dS analysis has mostly been used to look at evolution over a longer time period. However the analysis can also be used over a shorter time frame, as long as you don’t compare them to dN/dS ratios calculated over a longer time frame. In this study the time frame is the same in sampling 1 and sampling 2, and we therefore think it is scientific sound to compare them. 

8. You state the amino acid numbering scheme is based on the first methionine. In the flu literature H1 numbering occurs after the polypeptide sequence is cleaved, are you using this numbering scheme?

- International publications on H1 IAVs have both used numbering from the 1st methionine and the polypeptide sequence “DTIC”. I have now only included the DTIC numbering. 

9. The data would be highly strengthened if we were shown an amino acid alignment of the 2016 strain, the 2017 strain and the viruses sequences with difference in amino acids in the H1, if there are duplicate sequences they can be lumped, however Table 5 does not suffice for depicting the evolution. There isn’t even the known amino acid changes shown in “codon” in table 5.

- Another figure (Fig 3) containing an alignment of the vaccine strain H1av component, the enzootic screening H1av, the H1av of the 1st sampling and the H1av of the 2nd sampling has been added in the manuscript. This alignment also contains the antigenic sites. Moreover, the amino acid changes have been included in Table 5 and the specific antigenic sites are noted.

10. The strains used in the vaccine are known, please state them and speak to them and if the amino acid sequence is known, maybe even include it in the alignment (suggested in #8).

- The vaccine strains are now stated in the introduction (line 53-54) and the avian HA protein from one of the vaccine strains has also been included in the alignment with accession number (Fig 3). Furthermore, a section in the discussion has been added discussing the vaccine stains of the vaccine used in the study (line 590-595). 

11. All relevant viral segments for at least one represenative strain circulating on the farm in 2016 and in 2017 needs to be deposited into a proper publicly accessible database (NCBI) and referenced in the manuscript (A/Sw/xxx/2016 for example), this doesn’t need to be done with every isolate but virologists need a reference and you need to speak to that in alignments and phylogenetic trees to delineate the 2016 vs, 2017 outbreak.

- In the results section, the accession number of all sequences obtained during the study is given. In addition, I now refer to three representative strains of the study both with the sequence ID A/Sw/xxx/2016/2017 and accession number. 

12. In line with #10, the supplementary phylogenetic trees are suboptimal and provide minimal information as to the H1av diversity circulating in the country and how they related to the rest of Europe. You do not provide any real background information of the H1av subtype in the introduction and discussion. You mention your sequences on line 152 but what do they correspond to?

- It is not the meaning to show the H1av diversity within Denmark, but rather to illustrate the genetic diversity of the sequences obtained at the 1st and 2nd sampling round. Therefore the S1 and S2 figures are only cited in relation to the evolutionary analysis. The trees are molecular clock trees with a timeline below, which indicates how fast the viral strains evolved, therefore no other reference strains can be included in the trees. In the caption of the phylogenetic tree now include a statement that the names of the sequences observed in the tree are similar to the ID that the sequences are given by NCBI Genbank. A short description and an additional reference on the H1av-like viruses in Denmark have been included in the introduction.

13. Line 53, source #17, I did not find any data in this source that clarified that this vaccine provides protection to the Denmark subtype. While this is inferred from the H1avN1 component in the vaccine provides protection against your H1av do you have a proper source for this subtype?

- We acknowledge that this description was a bit misleading. There is no proper evidence that the vaccine covers all the Danish H1 variants, but the trivalent vaccine includes both one strain with a H1 gene and one strain with a N3 that belong to the linkage as the H1N2 virus. Therefore this vaccine is extensively used in Denmark to provide protection against the Danish H1N2 strain. However, we can discuss if the variations between the vaccine strains and the circulating strain have an impact on vaccine efficiency, but this should be tested in another setup. The sentence has now been revised and extended (line 53-57) 

14. Line 58- another unwanted effect reported by others in the swine influenza literature is vaccine associated enhanced respiratory disease (VAERD), please consider citing.

- VAERD has now been mentioned and cited in the introduction.

15. Line 24 abstract - state what the subtype is and provide a reference strain (see #10).

- This has now been added.

16. Figure 3 would be better resprented if a supplementary table showed the HI titers for each of the 20 pigs.

- This has now been implemented in S1 table. 

17. Figure 2 needs more descriptive labels to reflect before and after vaccination.

- More descriptive labels have now been added to Figure 2.

18. Figure 3, as already mentioned, many of your sows are sero-postive to flu, this data indicated that vaccination really didn’t do much. Also, did you sequence the P5-U4, HB4, and VB4 viruses before doing the HI? Were they grown in MDCK cells? Please include in materials and methods.

- As mentioned above the sero-positive sows at the time of vaccination is now discussed in line 529-537. 

- The viruses were grown and sequenced before doing HI-test, and this has now been emphasized in line 225-226. 

19. Fig. 3 is limiting as we have no idea about the antigenic cross-reactivity of sera to these H1av viruses. Any naiive serum as a control? Any positive swine serum you could obtain specific to the vaccine component from another group or previous study? it appears that all pigs have immunity to the 2016 strain (1st round- P5-U4), approximately half have immunity to the 2017 strain (HB-4) this highly impacts how the piglets might respond with these maternal antibodies.

- As stated in the materials and methods section, hyperimmune sera raised against the full vaccine and the H1av component of the vaccine were used as controls for the HI-test. Moreover, an H3N2 strain was used as control to ensure that the sows of the 2nd sampling indeed had been vaccinated. However, no naïve serum control was used. There is always a risk of cross-reactivity when performing HI-test. However, the sera obtained from the 1st and 2nd sampling round, were tested against the same strains, so if there was cross-reactivity it would be the case for all included samples, and therefore we can still observed differences between the two samplings. 

Reviewer #2: 

The major critiques of the manuscript are as follows:

1. Sample positivity is defined by qPCR Ct values <36. This is dependent on the amount of RNA utilized for the one-step qRT-PCR they employ in this manuscript. There is no mention of the amount of RNA from each sample that was used for the qRT-PCR assay. Please clarify that samples that were considered flu negative were not due to lack of sufficient RNA in the reaction. Please clearly state the amounts of RNA used for qRT-PCR.

- We did not measure the amount of RNA following extraction, and this has been emphasized in the material and methods (line 144-152) and is also mentioned in the discussion (line 510-512). 

2. Please show the multiple alignments of the original enzootic H1N1, HB4 and VB4 strain HAs as supplemental material.

- An alignment has now been included in the manuscripts as Fig 3. 

3. Line 337. please clarify what two-seven weeks (is it twenty seven, is it two through seven).

- This has not been corrected (374-375). 

4. In Table2, please provide range of the Ct values instead of the mean. This would help understand the spread of the data across each time point as well as compare run efficiencies.

- The range of Ct values at each sampling time has now been added. 

5. What Ct threshholds were utilized in the multiplexing assays used (line 309)?

- The threshold has now been stated in the materials and methods (line 151-152). 

6. Please speculate on why a majority of the amino acid changes in HA were located in the positions 91-307 between screening test and 1st sampling isolates. Do these changes improve viral fitness?

- This has now been emphasized in the discussion (line 578-580).

---

## [Editor Report · Decision Letter 1]

24 Oct 2019

Acute Influenza A virus outbreak in an enzootic infected sow herd: Impact on viral dynamics, genetic and antigenic variability and effect of maternally derived antibodies and vaccination

PONE-D-19-25024R1

Dear Dr. Ryt-Hansen,

We are pleased to inform you that your manuscript has been judged scientifically suitable for publication and will be formally accepted for publication once it complies with all outstanding technical requirements.

With kind regards,

Ralph A. Tripp

Academic Editor

PLOS ONE

Additional Editor Comments (optional):

The revised manuscript is now acceptable.
---

## [Editor Report · Acceptance letter]

1 Nov 2019

PONE-D-19-25024R1 

Acute Influenza A virus outbreak in an enzootic infected sow herd: Impact on viral dynamics, genetic and antigenic variability and effect of maternally derived antibodies and vaccination 

Dear Dr. Ryt-Hansen:

I am pleased to inform you that your manuscript has been deemed suitable for publication in PLOS ONE. Congratulations! Your manuscript is now with our production department. 

With kind regards,

on behalf of

Dr. Ralph A. Tripp 

Academic Editor

PLOS ONE